# Economic Value, Farmers Perception, and Strategic Development of Sorghum in Central Java and Yogyakarta, Indonesia

**Sugeng Widodo** [1,*], **Joko Triastono** [1], **Dewi Sahara** [1], **Arlyna Budi Pustika** [2], **Kristamtini** [2], **Heni Purwaningsih** [3], **Forita Dyah Arianti** [4], **Raden Heru Praptana** [2], **Anggi Sahru Romdon** [1], **Sutardi** [2], **Setyorini Widyayanti** [2], **Andi Yulyani Fadwiwati** [1] and **Muslimin** [1]

1   Research Center for Behavioral and Circular Economics, National Research and Innovation Agency, Jl. Jend. Gator Subroto No. 10, Jakarta 12710, Indonesia
2   Research Center for Food Crops, National Research and Innovation Agency, Jl. Raya Bogor-Jakarta, Cibinong Bogor 16911, Indonesia
3   Research Center for Food Technology and Processing, National Research and Innovation Agency, Jl. Jogja-Wonosari KM 31.5 Gading, Playen, Gunungkidul, Yogyakarta 55861, Indonesia
4   Research Center for Sustainable Production System and Life Cycle Assessment, National Research and Innovation Agency, Serpong, South Tanggerang City 15314, Indonesia
*   Correspondence: suge018@brin.go.id

**Abstract:** Sorghum is an important food crop commodity in the midst of climate change conditions and the threat of a global food crisis. Sorghum, which has an adaptive advantage to all land conditions, is suitable for use as a food substitute for rice and wheat. The purpose of this study was to evaluate the economic value, farmers' perceptions, and specific strategies for developing sorghum in Central Java and Yogyakarta, Indonesia. The research was conducted in Wonogiri Regency, Central Java, and Gunungkidul Regency, Yogyakarta from September to November 2022. The research was carried out through the observation of 120 respondents with indicators of farming characteristics and farmers' perceptions of sorghum development, as well as focus group discussions (FGD) and depth interviews with indicators of internal and external factors for sorghum development. The analysis used is benefit cost (BC) to evaluate the economic value of sorghum farming, the Likert scale to determine farmers' perceptions of sorghum, and Strength Weak Opportunity Threat (SWOT) to determine specific strategies for developing sorghum. The results showed that sorghum farming is feasible to develop in Wonogiri Central Java and Gunungkidul Yogyakarta because it provides a profit value greater than production costs with a BCR value of >1. The perception of farmers in Central Java regarding the development of sorghum is included in the very good category with an average value of 3.31, and the perception of farmers in Yogyakarta is included in the good category with an average value of 2.55. The operational policy strategy for developing sorghum in Wonogiri Central Java and Gunungkidul Yogyakarta is an expansion strategy (S-O).

**Keywords:** sorghum development; farm household economic; economic value; farmer perception; strategic policy

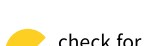

## 1. Introduction

Sorghum (*Sorghum bicolor* L. Moench) is a staple food for people around the world who live in semi-arid and subtropical countries in Asia and Africa [1]. Sorghum has anti-inflammatory and cholesterol-lowering properties so the consumption of sorghum as a food ingredient is increasing in high-income countries [2]. The high content of bioactive compounds in sorghum seeds and widespread public acceptance of sorghum as a breakfast cereal, beverage, and other products indicates a higher potential for sorghum consumption in the future in several countries such as the United States [3], Brazil [4], South Africa [5], and Kenya [6].

Sorghum has quite good market opportunities for the feed industry, alcohol refining, and flour; even now sorghum processing interventions create business opportunities for the food industry [7]. The prospects for the sorghum market are very promising by expanding the beverage industry, as well as creating jobs and markets for sorghum producers [8]. Sorghum has a high nutritional content so sorghum can replace rice as a food ingredient [9]. In Indonesia, sorghum is the third cereal food crop after rice and corn, but the use of sorghum as food has declined sharply after the availability of sufficient rice at a low price. In Indonesia, the development of sorghum was only seen in the 1940s as a source of food during the famine season. The sorghum planting area in 1990 reached 18,000 ha and was spread over the Demak and Wonogiri areas of Central Java, South Sulawesi, and East Nusa Tenggara. In 2020, the development of 5000 ha of sorghum occurred in East Nusa Tenggara, Yogyakarta, Southeast Sulawesi, and South Sulawesi [10]. Sorghum is a food source containing 332 calories, 73.0 g carbohydrates, 11.0 g protein, 3.3 g fat, 28 mg calcium, 287 mg phosphorus, 4.4 mg iron, and 0.38 mg vitamin B1 [11], so sorghum is feasible to be developed as an alternative food ingredient to replace rice.

Sorghum is a cereal crop that has great potential to be developed commercially because it has broad agro-ecological adaptability, high productivity, is relatively resistant to pests and diseases, does not require large inputs in cultivation, and is more tolerant of marginal conditions [12]. Sorghum, as a drought-tolerant crop that grows well on marginal lands, is a very important commodity that can serve as an alternative food and source of income for smallholder farmers [13]. The development of sorghum is still very slow because the popularity of sorghum is below that of maize [14]. At the farm level, sorghum is grown as an intercrop, a diversion crop for birds [11], and a side crop planted on the edge of the bunds without cultivation techniques so that productivity is still low [15].

Under conditions of global climate change, sorghum has the opportunity to become an important commodity as a food and industrial crop [16,17]. The Russo-Ukraine war affected the production countries to restrict wheat exports in order to prevent the food crisis, so the wheat-importing countries tried to optimize the local commodity resources that could replace the function of wheat. The Indonesian government has formulated a national sorghum development plan for 2022–2024 to increase production and downstream sorghum as a substitute for wheat to safeguard national food security from the threat of the global food crisis. Government efforts and policies are not sufficient only by doubling the area planted and increasing the productivity of sorghum, but also by creating markets and ensuring the level of absorption of sorghum in both domestic and foreign markets. In the road map for the national sorghum development program, it is scheduled that sorghum will be developed in 17 provinces with a target of 30,000 ha (2023) and 40,000 ha (2024) with a production target of 115,848 t (2023) and 154,464 t (2024) assuming productivity of 4.0 t ha$^{-1}$ [18].

Sorghum can be developed in all agricultural land agroecology in Indonesia. The extensification of sorghum cultivation on marginal land is the best alternative to increase the availability of carbohydrate sources. Sorghum plants can produce well on marginal land, so they do not reduce the area planted for rice and corn. In Indonesia, there are still around 60 million hectares of marginal land that have not been cultivated. Increasing sorghum production on marginal land is the most likely solution to reduce wheat imports [19]. Central Java and Yogyakarta have the potential for dry land which is distributed in several districts. Agricultural land in Wonogiri Regency, Central Java, which is dominated by dry land and rainfed paddy fields, has been partially utilized for the development of sorghum. In 2021, Wonogiri Regency will become one of the locations for the 50 ha sorghum development program from the Indonesian Ministry of Agriculture [20]. In Gunungkidul Regency, Yogyakarta, 90% of agricultural land is dry land [21], and sorghum has been developed in several areas. However, the development of sorghum in the region has not been managed upstream and downstream using the latest technology and business institutional governance has not been formed involving farmers, the government, and the private sector. Each area of sorghum development has different agroecological and

socioeconomic resources, so a different strategy is needed for developing sorghum in a region. This study aimed to evaluate the economic value, farmer perceptions, and specific strategies for developing sorghum in Central Java and Yogyakarta. The novelty value of this research is the area-specific strategy for developing sorghum based on the potential strength of regional resources, economic value, and farmers' perceptions as material for policy formulation for a large-scale sorghum development program.

## 2. Materials and Methods

### 2.1. Research Site

The research was conducted in Central Java (Wuryantoro and Pracimantoro Sub-Districts, Wonogiri Regency) and Yogyakarta (Wonosari and Karangmojo Sub-Districts, Gunungkidul Regency) from September to November 2022. The selection of the research location was carried out deliberately with the consideration that the location is a sorghum production center [22,23] and a target area for Indonesia's national sorghum development program for 2022–2024. Agricultural land in Wonogiri Regency is rainfed dry land and tides. In general, farmers own more than one plot of land planted with more than one crop commodity. Farmers cultivate crops in monoculture and intercropping. Working in the agricultural sector is very important because most of the family income comes from the agricultural sector [24]. Most of Gunungkidul is rainfed dry land which is very dependent on climate cycles and is the main food barn for the community. In general, farmers plant food crops twice a year. The choice of cropping pattern is a consideration for farmers in allocating labor because farmers work outside agriculture during the dry season to meet food and non-food needs. Farmer household behavior will allocate the income earned for food and non-food consumption needs so that they maximize their business with all the limitations they have [25]. Map of research activity locations in Central Java and Yogyakarta are presented in Figures 1 and 2.

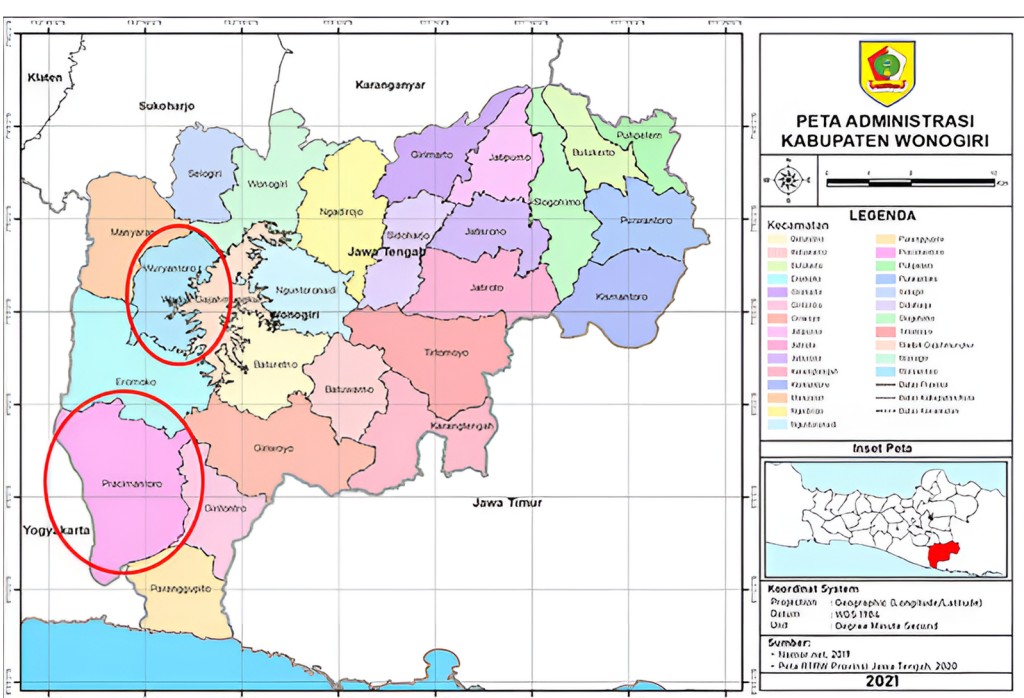

**Figure 1.** Wuryantoro and Pracimantoro Sud-District, Wonogiri Regency, Central Java. (Data from: Ahmad [26]).

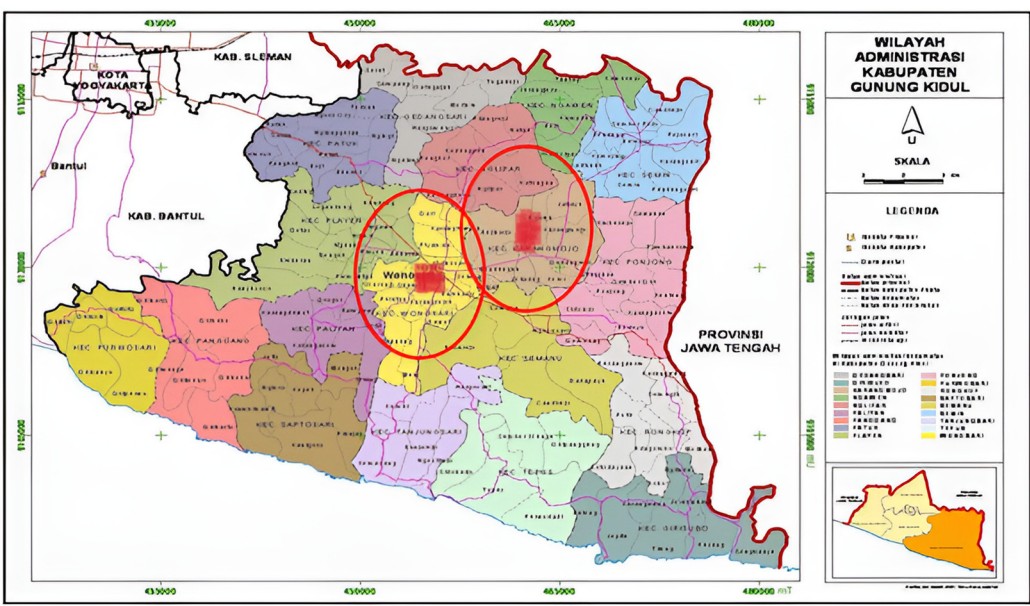

**Figure 2.** Wonosari and Karangmojo Sub-District, Gunungkidul Regency, Yogyakarta. (Data from: Peta Kota [27]).

## 2.2. Sampling Design

This study uses a cross-sectional survey design that collects data at a single point in time. The sampling of farmers was carried out in two stages; the first stage was selecting farmers in areas that were sorghum production centers, and the second stage was identifying farmers who planted sorghum in monoculture.

## 2.3. Data Collection

Primary data was collected using a survey method of 120 farmers who grow sorghum in selected locations (70 farmers in Central Java and 50 farmers in Yogyakarta) and FGD methods with farmers, traders, private actors (off-takers), and policyholders at the provincial/district level and home industry with a total 40 participants. The selection of respondents was carried out purposively, namely farmers who had cultivated sorghum in monoculture and obtained 70 respondents in Central Java and 50 respondents in Yogyakarta. The number of respondents in the two locations met the criteria for the number of respondents for survey research [28]. Each FGD participant conveyed information according to their field related to sorghum development: policyholder (support and sorghum development program); farmers (experiences, problems, and future hopes in developing sorghum); traders (market opportunities and purchasing capacity); off-takers (marketing, pricing, distribution, partnerships); and home industry (types of processed sorghum).

Primary data collection through interviews with farmers and traders who were selected as respondents using a list of questions. Primary data collected include the following: characteristics of sorghum farming; farmers' perceptions of sorghum development; as well as internal and external factors of the potential for sorghum development in Central Java and Yogyakarta which were collected through FGDs. Two topic sets were used in the interviews, namely the topic set to find out farmers' perceptions of sorghum development and the topic set about the financial feasibility of sorghum farming. The topic set for farmers' perceptions includes 11 indicators, namely the ease of obtaining sorghum seeds, seed growth, plant growth, plant maintenance, resistance to pests and diseases, sorghum production, sorghum market, sorghum marketing, sorghum seed processing, sorghum prices, and profits from sorghum farming. Topics set for financial feasibility include quantity and price of seeds, fertilizers, and pesticides, wages for labor, land tax, depreciation of agricultural equipment, production, and price of sorghum.

Secondary data were obtained from the Central Bureau of Statistics of Central Java and Yogyakarta Provinces, as well as some of the results of previous studies as listed in the Bibliography. Secondary data is used as information to support and discuss research results.

*2.4. Data Analysis*

2.4.1. Economic Value of Sorghum Farming

Evaluation of the economic value of sorghum developed by farmers in Central Java and Yogyakarta was analyzed by using the B/C approach of Yang et al. [29], as follows:

$$\pi_i = TR_i - TC_i \text{ and } B/C_i = \pi_i/TC_i$$

Description:
$\Pi_i$ = Farming profits *i*-th
$TR_i$ = Total farming revenue *i*-th
$TC_i$ = Total farming cost *i*-th
$B/C_i$ = Farming feasibility *i*-th

If the BCR value is 1.0, then farmers will benefit from sorghum farming, so they can continue sorghum development, whereas if the BCR value is <1.0, farmers will not benefit from sorghum farming and farming does not need to be continued [30].

2.4.2. Farmers' Perceptions of Sorghum Development

Data on farmers' perceptions of sorghum development in the form of ordinal data were analyzed using a scoring technique [31]. Farmers' perceptions were assessed using a Likert scale with a score of 1–4 in the very good, good, bad, and very bad categories. Furthermore, the perception data were analyzed using scoring with the formulation of Milkias et al. [32], as follows:

$$Nilai\ skore = \frac{n_i.s_i}{N_i}$$

Description:
$n_i$ = The number of respondents in the column *i*-th (*i* = 1, 2, 3)
$s_i$ = Statement score *i*-th (*i* = 1, 2, 3)
$N_i$ = The number of respondents on the row *i*-th (*i* = 1, 2, 3)

If the criterion value is between 1.00–1.75 = very bad perception category, 1.76–2.50 = bad perception category, 2.51–3.25 = good perception category, and 3.26–4.00 = very good perception category [32].

Farmers' perceptions of sorghum development are categorized by 3 class interval scales, namely high, medium, and low. The interval scale is determined by the following formula:

$$Interval\ Scale = \frac{The\ highest\ score - The\ lowest\ score}{Number\ of\ interval\ scale})$$

The diversity of farmer perceptions of sorghum development is visualized using the Perceptual Mapping technique which describes the relationship between farmer perceptions and predetermined attributes [33].

2.4.3. Strategic Development of Sorghum

Specific strategies for developing sorghum in Central Java and Yogyakarta were determined through a SWOT analysis. SWOT analysis begins with identifying the strengths (S) and weaknesses (W) in sorghum development, as well as opportunities (O) and threats (T) from the external environment that can maximize S and minimize W and T in sorghum development [34]. All of these factors were tabulated into the internal factor evaluation matrix (IFEM) and the external factor evaluation matrix (EFEM) and then given a weight rating and score for each factor's S, W, O, and T [35].

The SWOT analysis steps are as follows by LAN [36]:

- Identify internal factors in the form of S and W and external factors in the form of O and T.
- Determine the three priority factors of S, W, O, and T. The three priority factors of S, W, O, and T were determined based on the ranking of the choices of the FGD participants (40 people) with the following criteria: (1) the first priority is >50% of the participants; (2) second priority choice of 25–50% of participants; and (3) the third priority for <25% participants.
- Priority internal and external factors are then analyzed to determine the magnitude of the Urgency Value (UV). UV is the basis for determining the Factor Weight (FW) of each internal and external factor. UV value is determined by comparing the level of importance of one factor with other factors in the group of internal/external factors. UV ranges from 1 to 5 with the provision that the higher the UV value means the level of importance between one factor and another is very high and vice versa. BF is calculated by dividing the number of UV from each factor by the total value of the internal/external factor group and multiplied by 100.
- Determining the Key Success Factors (KSF) through evaluating internal and external factor linkages to determine the Support Value (SV) and the Support Weight Value (SWV), as well as the Average Linkage Value (ALV), the Linkage Weight Value (LWV) and the Total Weight Value (TWV) of each factor. The SV value is between 1 and 5 and the higher the SV value, the higher the support from that factor. The value of relatedness (VR) is determined by giving a score of 1 (very little relatedness) to 5 (very high relatedness). Key Success Factors is selected from the largest TWV from each of the factors of S, W, O, and T. Calculation of each factor analysis is as follows:

  SWV = FW × SV
  ALV = Total VR/n−1
  LWV = ALV × FW
  TWV = LWV + SWV

- Determine the strength map based on the results of the evaluation of the interrelationships of internal and external factors. The strength map is obtained by comparing the TWV from all S values with all W values and the TWV from all O values with all T values.
- Formulation of operational policy strategies using the SWOT strategy formulation. The four main strategies that can be formulated in the four SWOT quadrants are presented in Table 1:
- Preparation of activity plans by outlining each operational policy strategy in the form of activity plans that need to be implemented.

**Table 1.** SWOT strategy formulation.

| Quadrants | Strategy | Description |
| --- | --- | --- |
| 1 | Expansion Strategy (S-O) | Strategies use S to seize O |
| 2 | Diversification Strategy (S-T) | Strategies use S to overcome or minimize T |
| 3 | Stability or Rationalization Strategy (W-O) | Strategies to overcome W by taking advantage of O |
| 4 | Defensive or Survival Strategy (W-T) | The strategy of fixing W by minimizing T |

## 3. Results

### 3.1. Economic Value of Sorghum Farming

The production inputs used in sorghum farming in Central Java and Yogyakarta are relatively the same including seeds, fertilizers, pesticides, labor, and other fixed costs. Farmers in Wonogiri Regency used the red sorghum variety (Suri 3) with a seed amount of 13 kg ha$^{-1}$, and farmers in Yogyakarta used white sorghum (PB) and local red sorghum varieties with a seed amount of 10 kg ha$^{-1}$. The fertilizers used were urea, ZA, Phonska, and

different amounts of manure at the two locations, except that ZA was not used in Central Java. The total cost of sorghum production consists of variable costs (seeds, fertilizers, pesticides, and labor) and fixed costs (land tax and depreciation value of equipment), amounting to USD 382.08 and USD 672.68, respectively. The highest costs used in both locations were labor costs, which were USD 293.61 and USD 471.71 respectively. The sorghum production obtained in Central Java was USD 0,19 kg ha$^{-1}$ with a value of USD 808.64, whereas sorghum production in Yogyakarta was 5750 kg ha$^{-1}$ with a value of USD 1549.90. Analysis of the economic value of sorghum farming in Central Java and Yogyakarta is presented in Table 2.

**Table 2.** Economic value of sorghum farming in Wonogiri Central Java and Gunungkidul Yogyakarta, 2022.

| Types of Production Input | Central Java | | Yogyakarta | |
|---|---|---|---|---|
| | Physical (ha$^{-1}$) | USD (ha$^{-1}$) | Physical (ha$^{-1}$) | USD (ha$^{-1}$) |
| Variable Cost: | | | | |
| a. Seed (kg) | 13 | 3.23 | 10 | 3.85 |
| b. Fertilizer (kg) | | | | |
| • Urea | 110 | 16.94 | 150 | 23.10 |
| • ZA | - | | 230 | 36.90 |
| • Phonska | 110 | 17.65 | 280 | 50.32 |
| • Manure | 870 | 19.51 | 2250 | 50.54 |
| c. Pesticides (package) | | 12.84 | | 21.82 |
| d. Labor (man days) | 65 | 293.61 | 90 | 471.71 |
| Total Variable Cost | | 363.79 | | 654.39 |
| Fixed Cost | | | | |
| a. Land Tax | | 7.06 | | 7.06 |
| b. Equipment Depreciation Costs | | 11.23 | | 11.23 |
| Total Fixed Cost | | 18.29 | | 18.29 |
| Total Farming Cost | | **382.08** | | **672.68** |
| Production (kg) | 3.000 | 808.64 | 5750 | 1549.90 |
| Benefits | | 436.10 | | 877.22 |
| BCR | | 1.14 | | 1.30 |

Noted: Source = Primary Data 2022 (processed); 1 USD = 15,581.65 IDR (23 December 2022).

*3.2. Farmers' Perceptions*

The survey results show that 63.64% of farmers in Central Java have a very good perception and 36.36% of farmers have a good perception of the 11 indicators of sorghum development. Meanwhile, of farmers in Yogyakarta, as much as 54.55% have a good perception, 36.36% not good, and 9.09% very bad. Overall, the perception of farmers in Central Java towards the development of sorghum is included in the very good criteria with an average value of 3.31, and the perception of farmers in Yogyakarta is included in the good category with an average value of 2.55. Farmers' perceptions of the sorghum price indicator in Yogyakarta are very unfavorable. Farmers' perceptions of sorghum development in Central Java and Yogyakarta is presented in Figure 3.

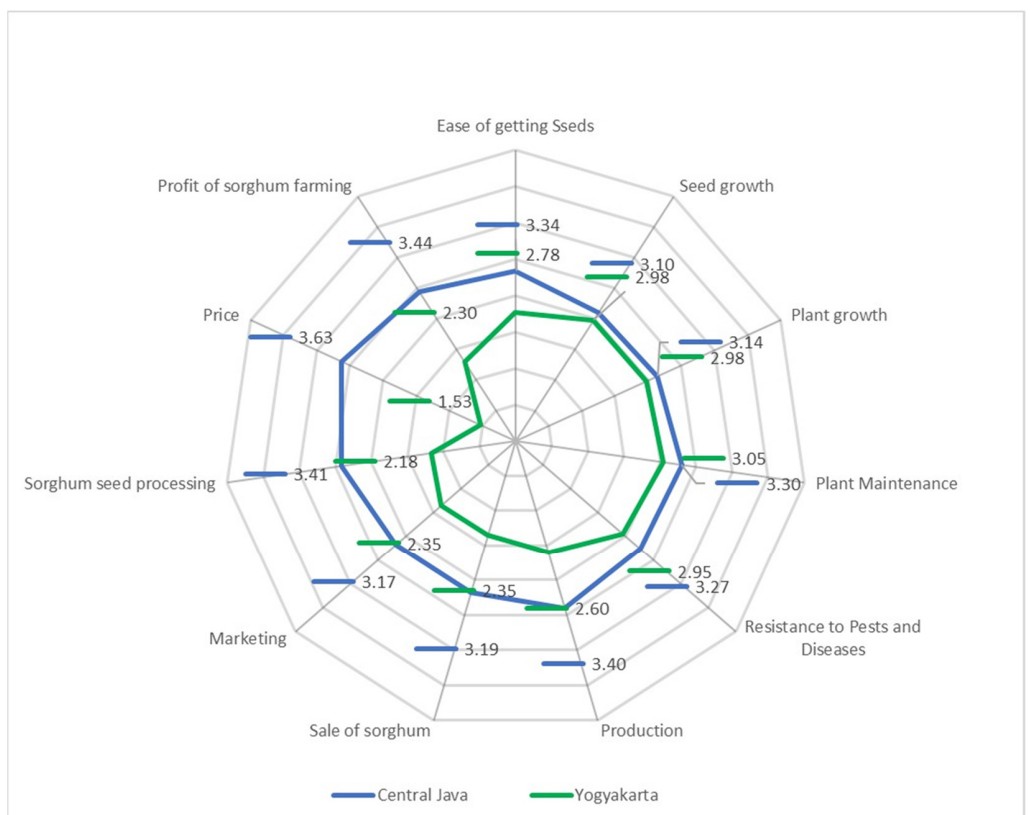

**Figure 3.** Farmers' perceptions of sorghum development in Central Java and Yogyakarta, 2022. Source = Primary Data 2022 (processed).

*3.3. Strategic Development of Sorghum*

3.3.1. Internal, External, and Priority Key Factors

Through FGD, internal factors (S and W) and external factors (O and T) which are priority factors in sorghum development in Central Java and Yogyakarta have been identified. Inside the internal factors, there are two factors that are the same as S for sorghum development in Central Java and Yogyakarta, namely agro-climatic and land availability, although they differ in the priority ranking. Likewise, with W, there are also two common factors, namely cultivation technology that is not yet intensive and prices that are not yet stable. Meanwhile, in terms of external factors, the high demand for sorghum is an O in both regions. All the factors that pose a T to sorghum development in the two regions are the same, namely pest attacks, land competition, and climate anomalies. The identification results of three priority internal factors and three priority external factors for sorghum development in Central Java and Yogyakarta are presented in Tables 3 and 4.

The results of the KSF evaluation of sorghum development in Central Java showed that the S factor was a favorable agro-climatic (BNP = 1.42); the W factor is the unstable price (TWV = 0.63); the O factor is the high demand for sorghum processed products and products (TWV = 0.93); and the T factor is pest attack (TWV = 0.41). Meanwhile, the results of the KSF evaluation for sorghum development in Yogyakarta showed that the S factors are as follows: land potential in the third planting season and unused land (BNP = 2.18); cultivation is not yet intensive/traditional (BNP = 1.55); market demand is high, and prices are starting to improve (TWV = 2.25); and the T factor is competitive land use (BNP = 0.87). The results of evaluating internal and external factors on sorghum development in Wonogiri Central Java and Gunungkidul Yogyakarta are presented in Tables 5 and 6.

**Table 3.** Priority factors of SWOT to sorghum development in Wonogiri Central Java, 2022.

| Priority | S | W | O | T |
|---|---|---|---|---|
| 1 | Agro-climatic supports (60%) | Unstable prices (55%) | Demand for sorghum processed products and products is high (60%) | Pest attack (62.5%) |
| 2 | Low input/low production costs (25%) | Cultivation and processing technology has not been mastered (27.5%) | Utilization of waste/biomass has a high added value (30%) | Competitive in land use especially with corn commodity (25%) |
| 3 | Available land that can be utilized in the third growing season (15%) | There is no pre and post-harvest mechanization available (17.5%) | There is no sorghum seed cultivator yet (10%) | Climate anomaly, if during the third growing season (on season) it rains a lot, farmers plant other commodities (12.5%) |

Noted: Source = Primary Data 2022 (processed); Numbers in brackets indicate the percentage of FGD participants who chose priority factors.

**Table 4.** Priority factors of SWOT to sorghum development in Gunungkidul Yogyakarta, 2022.

| Priority | S | W | O | T |
|---|---|---|---|---|
| 1 | Land potential in the third growing season and unused land (52.5%) | Cultivation is not yet intensive/traditional (55%) | Market demand is high, and prices are starting to improve (62.5%) | Climate anomaly/seasonal shift (65%) |
| 2 | Human resources are available and it is customary to grow sorghum (27.5%) | Productivity of sorghum seeds is still low, more dominant for animal feed (25%) | There began to be off-takers/exporters specifically for certain/local red sorghum varieties (27.5%) | Pests (birds, whitefly, long-tailed monkeys, rats) (25%) |
| 3 | Land suitability and climate support (appropriate agro-ecosystem) (20%) | Prices for dry beans are still low (20%) | Local food diversification: rice, flour, tempeh, and sorghum added value is starting to improve (10%) | Competitive land use (10%) |

Noted: Source = Primary Data 2022 (processed); Numbers in brackets indicate the percentage of FGD participants who chose priority factors.

**Table 5.** Evaluation of internal and external factors in sorghum development in Wonogiri Central Java, 2022.

| Internal and External Factors | FW (%) | SV | SWV | ALV | LWV | TWV | KSF |
|---|---|---|---|---|---|---|---|
| **S** | | | | | | **2.14** | |
| Agro-climatic supports | 26.67 | 5 | 1.33 | 4.00 | 1.07 | 1.42 | 1 |
| Low input/low production costs | 20.00 | 5 | 1.00 | 3.27 | 0.65 | 0.65 | 2 |
| Available land can be utilized in the third growing season | 6.67 | 5 | 0.33 | 2.82 | 0.19 | 0.06 | 3 |
| **W** | | | | | | **1.02** | |
| Unstable prices | 20.00 | 4 | 0.80 | 3.91 | 0.78 | 0.63 | 1 |
| Cultivation and processing technology has not been mastered | 20.00 | 3 | 0.60 | 3.00 | 0.60 | 0.36 | 2 |
| There is no pre- and post-harvest mechanization available | 6.67 | 3 | 0.20 | 2.36 | 0.16 | 0.03 | 3 |
| **O** | | | | | | **1.46** | |
| Demand for sorghum processed products and products is high | 26.67 | 4 | 1.07 | 3.27 | 0.87 | 0.93 | 1 |
| Utilization of waste/biomass has a high added value | 13.33 | 5 | 0.67 | 3.36 | 0.45 | 0.30 | 2 |
| There is no sorghum seed cultivator yet | 13.33 | 4 | 0.53 | 3.27 | 0.44 | 0.23 | 3 |
| **T** | | | | | | **0.52** | |
| Pest attack | 26.67 | 2 | 0.80 | 2.91 | 0.78 | 0.41 | 1 |
| Competitive land use especially with maize | 13.13 | 2 | 0.27 | 2.91 | 0.39 | 0.10 | 2 |
| Climatic anomaly, if during the third growing season (on season) it rains a lot, farmers plant other commodities | 6.67 | 1 | 0.07 | 1.55 | 0.10 | 0.01 | 3 |

Noted: Source = Primary Data 2022 (processed).

**Table 6.** Evaluation of internal and external factors in sorghum development in Gunungkidul Yogyakarta, 2022.

| Internal and External Factors | FW (%) | SV | SWV | ALV | LWV | TWV | KSF |
|---|---|---|---|---|---|---|---|
| **S** | | | | | | **5.01** | |
| Land potential in the third growing season and the land has not been utilized | 26.67 | 5 | 1.33 | 3.18 | 0.85 | 2.18 | 1 |
| Farmers' resources are available and they usually grow sorghum | 20.00 | 5 | 1.00 | 3.55 | 0.71 | 1.71 | 2 |
| Land suitability and supportive climate (agro-ecosystem) | 13.33 | 5 | 0.67 | 3.36 | 0.45 | 1.12 | 3 |
| **W** | | | | | | **3.33** | |
| Cultivation is not yet intensive/traditional | 20.00 | 4 | 0.80 | 3.73 | 0.75 | 1.55 | 1 |
| Productivity is still low, dominant for animal feed | 13.33 | 5 | 0.67 | 3.09 | 0.41 | 1.08 | 2 |
| Prices for dry beans are still low | 13.33 | 3 | 0.40 | 2.27 | 0.30 | 0.70 | 3 |
| **O** | | | | | | **4.48** | |
| Market demand is high, and prices are starting to improve | 26.67 | 5 | 1.33 | 3.45 | 0.92 | 2.25 | 1 |
| Start there off-taker/exporter red local variety and new superior varieties | 20.00 | 3 | 0.60 | 3.36 | 0.67 | 1.27 | 2 |
| Local food diversification: rice, flour, sorghum tempeh | 13.33 | 4 | 0.53 | 3.18 | 0.42 | 0.96 | 3 |
| **T** | | | | | | **2.06** | |
| Competitive land use | 13.33 | 3 | 0.40 | 3.55 | 0.47 | 0.87 | 1 |
| Climate anomaly/seasonal shift | 13.33 | 2 | 0.27 | 3.00 | 0.40 | 0.67 | 2 |
| Pests (birds, whitefly, long-tailed monkeys, rats) | 13.33 | 2 | 0.27 | 1.91 | 0.25 | 0.52 | 3 |

Noted: Source = Primary Data 2022 (processed).

### 3.3.2. The Strength Maps

The results of evaluating the relationship between internal and external factors in the development of sorghum in Central Java show that the position of the strength map is as follows: S of 2.14; W of 1.02; O of 1.46; and T of 0.52. After the TWV value of all S values is reduced by all W values, a value of 1.12 is obtained, and the TWV value of all O values is reduced by all T values, which is 0.94. Meanwhile, the position of the strength map of sorghum development in Yogyakarta is S of 5.01; W of 3.33; O of 4.48; and T of 2.06. The TWV value of S minus W is 1.68, and the value of TWV O minus T is 2.42. The strength map of sorghum development in both Central Java and Yogyakarta is in quadrant I or S-O with the proportion of S being greater than O. The priority strategy for developing sorghum is an expansion strategy (S-O), namely a strategy to utilize S to seize O. A strength map of sorghum development in Wonogiri Central Java is presented in Figure 4, whereas a strength map of sorghum development in Gunungkidul Yogyakarta is presented in Figure 5.

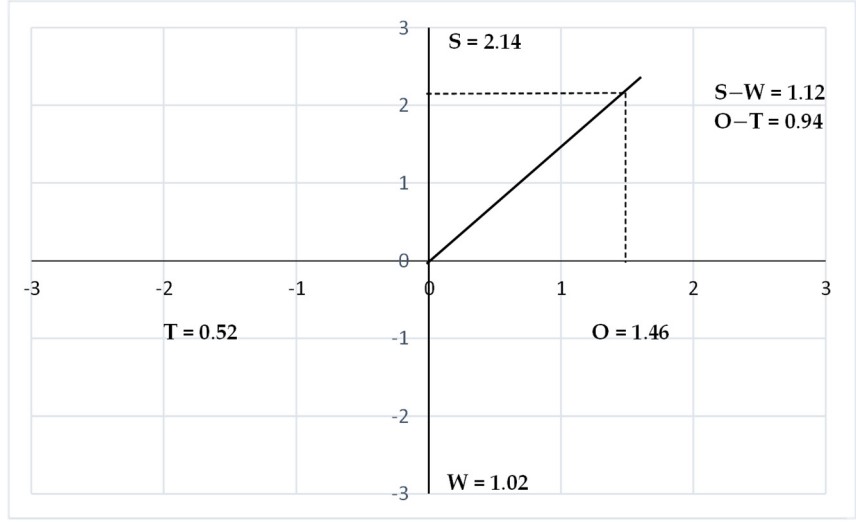

**Figure 4.** Map of the strength of sorghum development in Wonogiri Central Java, 2022.

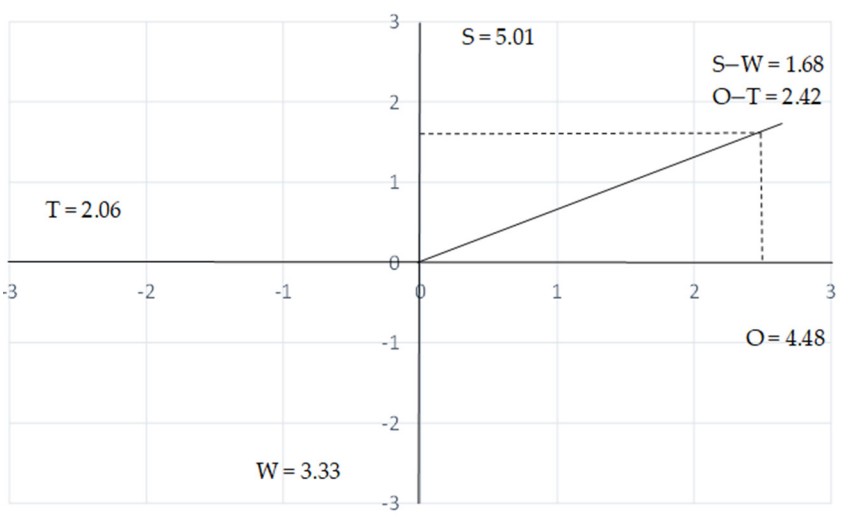

**Figure 5.** Map of the strength of sorghum development in Gunungkidul Yogyakarta, 2022.

## 4. Discussion

### 4.1. Economic Value of Sorghum Farming

Farmers in Wonogiri use more seeds than farmers in Yogyakarta, even though the amount of seed used is in accordance with the seed requirements for sorghum production, namely 10–15 kg ha$^{-1}$. The use of more seeds in Central Java is due to a distance planting, namely 20 cm × 60 cm or 40 cm × 40 cm, whereas the distance planting used in Yogyakarta is 30 cm × 50 cm or 40 cm × 70 cm. The closer the distance planting and the greater the number of seeds per hole, the more seeds are needed. The use of fertilizers in the two locations is still not according to the recommendations, but is performed according to the habits and capital capabilities of the farmers. Farmers with low incomes are unable to use fertilizer according to plant needs [37]. Sorghum farmers in Nigeria use fertilizers under recommendations because the availability of government-subsidized fertilizers is not easily accessible [38].

The variable cost of sorghum production in both locations ranged from 95.21–97.73% of the total production cost, and the highest cost was used to pay for labor, which was 70.12–76.85% of the total production cost. This is in line with the labor cost for sorghum production in India which is 42.46% [39] and 54% of the total production cost [40], whereas other costs have a lower proportion. The lower number of workers used in Central Java indicates a farming system that is not yet intensive, characterized by minimum tillage without even tillage, and rarely monitoring seed germination after planting. The use of labor in Yogyakarta is equivalent to the use of labor in sorghum farming in India, an average of 84.64 man-days [39].

The profits received by farmers in Yogyakarta are higher than those received by farmers in Central Java. Based on the BCR value >1, sorghum farming in both locations is feasible to develop because it provides a profit value greater than production costs. This value is higher than the BCR value of sorghum farming in Sinai Peninsula, Egypt of 0.65 [41], and Nigeria of 0.91 [42]. This indicates that farmers benefit from sorghum farming with varying values due to differences in the amount and cost of production.

### 4.2. Farmers' Perceptions

In general, farmers in Central Java have a better perception of sorghum development than farmers in Yogyakarta. A stark contrast can be seen in the Yogyakarta farmers' perception of the price of sorghum which is not very good (1.53). Farmers in Yogyakarta have experienced very unfavorable sorghum prices between USD 0.06–0.11 kg$^{-1}$, whereas farmers in Central Java have very good perceptions of sorghum prices because farmers have obtained sorghum prices between USD 0.20–0.32 kg$^{-1}$. Even so, until now the price

of sorghum is still unstable and the government has not set a price for sorghum in the same way as other food (rice) prices. The perception of farmers in Yogyakarta with unfavorable criteria can also be seen in the indicators of sorghum processing (2.18), profits (2.30), sales, and marketing of sorghum, each indicator having a value of 2.35. The unfavorable criteria for the four indicators are due to the absence of sorghum seed processing facilities and market access which is still difficult so farmers are worried that production will not be absorbed.

Farmers have good perceptions of cultivation techniques because farmers easily obtain seeds, grow seeds and plants easily and quickly, are relatively resistant to pests and diseases, have high crop production, and can be harvested more than once and provide benefits. Mukin et al. [43] state that the sorghum cultivation technique is very easy for farmers to do and harvesting sorghum can be completed two to three times a season.

In contrast to farmers in Yogyakarta, farmers in Central Java have good perceptions of indicators of seed growth, plant growth, sales, and marketing of sorghum, whereas very good perceptions are obtained on indicators of ease of obtaining seeds, cultivation techniques, sorghum processing, price, and profit. Farmers in several areas also have good perceptions of sorghum cultivation techniques. Farmers in Bantul Regency show a good perception of the development of sorghum plants through cultivation innovation and utilization of sorghum [44]. Farmers in East Wellega and West Shewa in Oromia State, Ethiopia showed a good perception of sorghum crops that have a tolerance to bird attacks [45]. Farmers in Lembata Regency have a very good perception of the development of sorghum farming [46].

Sales and marketing of sorghum in Central Java are perceived as very good because there are sorghum markets or traders so farmers have no difficulty selling sorghum seeds. Sales of sorghum is an indicator that is very important to less important depending on different ethnic groups [47]. Likewise, the price of sorghum and the benefits obtained are perceived very well by farmers because the average price of sorghum is USD 0.27 kg$^{-1}$ which is almost equivalent to the price of harvested dry grain. If rice is the main crop cultivated by farmers, sorghum is a secondary crop grown by farmers with less intensive business, so if the price of sorghum seeds is close to the price of grain, farmers will respond positively. High sales of sorghum will affect the amount of profit farmers receive. Nevertheless, farmers hope that the government can maintain sorghum price stability and market certainty to accommodate the sorghum produced by farmers.

### 4.3. Strategic Development of Sorghum

4.3.1. Priority Internal Factors for Sorghum Development

*Strength*

a. Agro-climatic conditions

The agro-climatic conditions in the Wuryantoro and Pracimantoro sub-districts of Wonogiri support the growth of sorghum plants [47]. Sorghum has been grown by the local community for generations on a small scale using conventional cultivation techniques until the last two years it began to be developed intensively in paddy fields and dry land. The agro-climatic conditions in the Karangmojo and Wonosari sub-district of Gunungkidul are also suitable for the growth of sorghum plants. The sub-districts of Semin, Ponjong, Rongkop, Semanu, Tepus, and Girisubo have potential criteria for growing sorghum [48].

The environment plays an important role in sorghum production [49]. Agroecological conditions are a prerequisite for increasing sorghum production in irrigated land in the Republic of Mexico [50]. Sorghum plants can grow well at temperatures of 23–30 °C, and relative humidity of 20–40% with altitudes above 500 m asl. Sorghum can grow well in almost all types of soil with a pH ranging from 5.0 to 7.5 [51]. The research results of Juniarti et al. [52] showed that land in Padang Laweh District, Sijunjung Regency, West Sumatra has the potential for the development of sorghum plants with characteristics of an average temperature of 25–27 °C, rainfall <200 mm, humidity <75%, good drainage, soil depth >60 cm, pH 4.4–6.1, and low availability of nutrients N, P, and K. Soil characteristics,

climate, and topography are the main criteria used to determine land suitability for sorghum cultivation in the Agamsa sub-catchment of Northeast Ethiopia [53]. The main factors affecting the suitability of land for growing sorghum in the Jinsha River basin are the slope of the terrain, the height, and the thickness of the soil layer. Factors such as soil texture, the certainty of water sources, and drainage conditions also have an impact on land suitability for sorghum crops [54].

b.    Low production costs

The low cost of production inputs is the reason for farmers in Wonogiri to plant sorghum in the third growing season, both on rainfed and dry land. Adaptability is quite good on dry land and grows well with minimal input, causing sorghum to grow widely. Due to its high production potential and low input use, sorghum is cultivated in tropical, subtropical, and temperate regions of the warmer semi-arid regions of the world [1]. Tesema [55] states that the main inputs for sorghum production are seeds, soil, and fertilizer. Sorghum production can be increased by 26% without increasing the input of land area and labor allocation, even input costs can be reduced by 56% without changing production levels [56].

c.    Availability of land

The availability of land in the Wuryantoro and Pracimantoro Wonogiri sub-districts is quite large for sorghum development, both intensive paddy fields and dry land. The availability of land in the Gunungkidul area is considered quite extensive, especially in dry land or rainfed paddy fields. During the third planting season from May/June to August/September, there is still a lot of unused land and only part of the land is used for planting sorghum or other crops.

Soil physical and chemical characteristics are the most important parameters in determining land suitability for sorghum development among parameters of climate, topography, and land erosion rate [57]. Al-Mashreki et al. [58] used an overlay technique to determine land suitability for sorghum development based on scoring four agro-climatic factors, namely soil, climate, erosion level, and topography. Based on the physical and chemical analysis of the soil, land suitability classes can be distinguished into two classes, namely actual and potential land for the development of sorghum crops [59]. Actual land means that the physical and chemical properties of the soil are suitable for optimal growth of sorghum plants, whereas on potential land there are still limiting factors that can still be improved [60]. Potential land can be improved by adding organic matter, NPK fertilizer, terracing, planting cover crops, and drying [61]. Suitable land for the development of sorghum in the Wuryantoro Wonogiri District area includes Wuryantoro, Genukharjo, Gumiwanglor, Mlopoharjo, Pulutankulon, Pulutanwetan, Mojopuro, and Sumberejo [47]. Potential land suitable for sorghum development in Gunungkidul is evenly distributed in the sub-districts of Wonosari, Karangmojo, Ponjong, Tepus, and Girisoba; even in the Wonosari and Karangmojo sub-districts, there is 8898 ha of land that can be used for sorghum development. This is supported by cattle and goat livestock centers which require animal feed from sorghum waste.

d.    Human resources are available and it is customary to grow sorghum

Sorghum has long been cultivated in Gunungkidul and developed in the 1980s, but its utilization is more dominant for animal feed. Utilization of land area, seeds and manure have a positive effect on sorghum production in Gunungkidul [62]. Availability of land and demand for animal feed causes farmers in Gunungkidul to develop sorghum. Sorghum farming contributes 2% to farmers' total income [14]. Farmers gain experience cultivating sorghum from generation to generation, making it easier to accept technological advances to increase production.

*Weaknesses*

a.   Prices are unstable and tend to be low

The price of sorghum in 2022 is considered by farmers to be quite good, between USD 0.26–0.28 kg$^{-1}$, which was previously only around USD 0.06–0.11 kg$^{-1}$. The price of dry sorghum seeds tends to be unstable, influenced by the quality and quantity of sorghum seeds. Apart from the quality of sorghum, limited sorghum traders, price guarantee, and the absence of off-takers, farmers consider it more profitable if it is used for animal feed, so farmers grow sorghum only as a border and are not managed intensively. Sapanali et al. [63] state that inappropriate selling prices are a farming risk. Farmers hope that the price of sorghum will be as stable as it is today and that there will be markets and home industries that can absorb the sorghum yields.

b.   Farmers have not mastered cultivation and processing technology

In Wonogiri, generally, farmers do not apply intensive cultivation techniques in growing sorghum so the resulting production is not optimal between 2.4–3.0 t ha$^{-1}$. Farmers also have not processed sorghum into semi-finished products (analog rice or flour) or finished products (various foods or beverages). Sorghum cultivation in Gunungkidul is still traditional because farmers perceive the sorghum business as a side business to fill fallow or intercropping land. Farmers planted sorghum as a border, planted with an intercropping system, without providing fertilizer and without controlling pests and diseases. Farmers also do not maintain ratoons properly, so the productivity of the ratoons is very low and they are only used for animal feed. Increased sorghum production must be supported by technology and capacity building of farmers in the production and post-harvest processes. The added value of processing a commodity can provide better prices for farmers [63,64].

c.   There is no pre-harvest and post-harvest mechanization available

The development of sorghum in Wonogiri has not been supported by the availability of agricultural mechanization. In Wuryantoro District, farmers use a rice threshing machine to separate the sorghum seeds from the panicles, whereas in Pracimantoro this is performed manually. Machines for processing sorghum seeds into rice and flour are available outside the region at quite high prices, namely USD 0.95 kg$^{-1}$. Agricultural machinery plays a major role in farming, especially in times of labor shortages. The use of agricultural mechanization in a fairly wide area provides several benefits in the form of saving time, reducing labor use, reducing costs, increasing productivity, and reducing yield losses [65].

d.   The productivity of sorghum seeds is still low

The productivity of sorghum in Gunungkidul is still low, ranging from 1–2 t ha$^{-1}$, but there is additional income indirectly from utilizing sorghum waste for animal feed. The low productivity of sorghum is because farmers have not utilized technology, especially the use of superior varieties, and plant maintenance is not carried out intensively, even without fertilizer application. Rahman et al. [66] stated that the application of cultivation technology, especially the use of high-yielding sorghum varieties, was able to increase production.

4.3.2. Priority External Factors for Sorghum Development

*Opportunities*

a.   Demand for sorghum processed products and products is high

Sorghum is one of the potential commodities that can be developed to support food and energy diversification programs in Indonesia. Sorghum has good market prospects as an industrial raw material to meet consumer demand for food diversification. the use of sorghum in the industry has a fairly good market opportunity to expand the industry and create jobs [7,8]. Currently, the demand for sorghum seeds is quite high with prices up to USD 0.38 kg$^{-1}$. The utilization of sorghum seeds is mostly for food, especially rice, flour, bird feed, and mixed animal feed. A number of off-takers have started looking for sorghum raw materials, but not much is known about the sorghum-producing centers.

b.    Waste and biomass have added value

The utilization of sorghum plants as animal feed has very open opportunities [67]. Several varieties of sorghum contain nutrients suitable for forages such as Keller and Wray [68]. The Pahat variety of sorghum was able to provide a total biomass yield potential of 26.6 t ha$^{-1}$, with a crude protein content of 10.95%, 92.23% organic matter, and 58.77% NDF [69]. The relatively large yield potential and high nutrient content can meet the nutritional needs of ruminants, so sorghum can be a source of feed both now and in the future.

c.    There is no sorghum seed cultivator available

Seed is the main input in crop production through its physical, physiological, and genetic qualities which affect plant growth and development [70]. Production of quality seeds is an opportunity for farmers to improve the quality of sorghum seeds [71]. Most of the sorghum seeds in Wonogiri come from their own harvest and buy seeds from shops or from government programs because there are no sorghum seed growers.

d.    Starting to have off-takers

One of the keys to the development of sorghum is the existence of a partner who guarantees price and production. Farmers will use their land to develop sorghum if there is a market available to accommodate the harvest. The market is the main factor for broadly developing sorghum [9]. Until 2021 there have been no off-takers who have collaborated with sorghum farmers in Gunungkidul. Currently, there are pilot collaborations with local off-takers to utilize local red varieties of sorghum seeds as a food mixture.

e.    Local food diversification

In Gunungkidul, sorghum has begun to be processed into sorghum rice, sorghum flour, and several other products such as sorghum snacks and sorghum tempeh. However, several types of processed products are still on a limited scale and are marketed in the Gunungkidul area. The utilization of sorghum in the form of flour is more profitable because it is more practical and easier to process into various snack products. Sorghum flour can be used as a raw material for making various types of snacks. Sorghum flour has a fine texture and the amino acids that make up its protein are able to form gluten better than corn flour, although quantitatively and qualitatively it is lower than wheat flour. Sorghum flour can substitute up to 80% of flour for pastries (cookies), 40–50% for cakes, 30–35% for noodles, and 15–20% for bread and the like without significantly reducing the taste, texture, and aroma [10].

*Threats*

a.    Pest attack

The dominant pests in the development of sorghum in Wonogiri and Gunungkidul are birds. Bird pests are one type of pest that causes the highest crop failure in sorghum plants [72]. This pest can cause up to 100% yield loss if there are no rice or corn crops at harvest. Birds eat sorghum seeds, especially the white ones and the relatively low tannin content [73]. A bird weighing 40–50 g is able to consume 10 g day$^{-1}$ of sorghum seeds [10]. Farmers in Gunungkidul save sorghum panicles using net traps, wrapped in plastic bottles. Some of the things that affect the high rate of bird attacks are the location of the sorghum plants which are close to bird breeding habitats, planting that is not simultaneous, and varieties. Methods of controlling birds include planting simultaneously in large expanses, installing bird scare devices, and environmental sanitation from weeds which are the habitat of birds [74].

b.    Competitive land use with corn commodity

Seeing the potential of sorghum which has wide adaptability in various types of land, it is feared that there will be competition for land use with other palm crops such as corn. Generally, sorghum is planted after the rice harvest if it is developed in intensive land. If

the price of sorghum is the same or higher than corn and there is a guaranteed market, farmers will tend to use their land to plant sorghum because it is easier to cultivate and requires lower production inputs.

c.    Climate anomaly

Climate anomaly is a shift of the seasons from the normal average. Climate anomalies such as the occurrence of La Nina affect the development of sorghum. Climate shifts caused sorghum cultivation in Gunungkidul to experience obstacles during harvesting and processing. Farmers experienced difficulties in drying during the sorghum harvest in August–September 2022 when it still rained. Another impact is a shift in the planting season and changes in cropping patterns and systems. Climate change causes a shift in the start of the rainy season and dry season which causes a change in the planting season [75]. In Wonogiri, sorghum is generally planted during the third growing season or the dry season. Wet climate conditions make farmers switch to more profitable commodities such as corn, peanuts, or rice. The rice–rice–corn cropping pattern is more profitable than other cropping patterns in a La Nina climate [76]. Changes in rainfall cause a reduction in paddy fields, changes in river and groundwater discharge, decreased productivity, decreased planting area and harvested area, decreased yield quality, decreased cropping index, and increased pest attacks [77].

### 4.3.3. Sorghum Development Policy Strategy

Based on the power map of sorghum development in Central Java and Yogyakarta, a SWOT strategy formulation can be formulated in the form of a SWOT matrix which is formulated from the four key success factors. The formulation of the SWOT strategy for developing sorghum in Central Java and Yogyakarta is presented in Tables 7 and 8.

**Table 7.** SWOT strategy formulation for sorghum development in Wonogiri Central Java, 2022.

| Factor Internals / Factor External | S Agro-Climate Supports | W Unstable Prices |
|---|---|---|
| O Demand for sorghum processed products and products is high | **S-O** Increasing productivity in accordance with regional agro-climatic conditions to meet the high demand for sorghum-processed products and products | **W-O** Collaboration with off-takers related to guaranteeing reasonable prices to increase production to meet the high demand for sorghum-processed products and products |
| T Pest attacks | **S-T** Optimizing cultivation techniques with innovative technology according to regional agro-climatic conditions to overcome pest attacks | **W-T** Collaboration with off-takers regarding guaranteed reasonable prices to increase income so that farmers have the financial ability to deal with pest attacks |

**Table 8.** SWOT strategy formulation for sorghum development in Gunungkidul Yogyakarta, 2022.

| Factor Internals / Factor External | S Land potential in the Third Growing Season and Unused Land | W Cultivation Is Not Yet Intensive/Traditional |
|---|---|---|
| O Market demand is high, and prices are starting to improve | **S-O** Take advantage of the land's potential with technology in increasing productivity and product quality to meet high market demand and prices that are starting to improve | **W-O** Improving more intensive cultivation of sorghum to meet market demands and improve prices |
| T Competitive land use | **S-T** Optimizing land in the third growing season and other fields to reduce competition in land use for other food crops | **W-T** Improving sorghum cultivation to reduce land use competition |

To operationalize the strategy that has been formulated through the formulation of the SWOT strategy, each strategy is further translated into action plans that need to be implemented. The activity plan implemented in Central Java is different from that in Yogyakarta according to each operational policy strategy that is formulated. This operational policy strategy serves as a reference for policymakers in preparing sorghum development programs broadly in their respective regions. However, the results of this study have not shown the efficiency of sorghum production and sorghum development business institutions, so a more comprehensive study is needed on sorghum production efficiency by implementing sorghum cultivation standard operational procedures (SOPs) and a number of dimensions based on proper governance to establish sorghum development business institutions. The operational policy strategy for developing sorghum in Central Java and Yogyakarta is presented in Tables 9 and 10.

**Table 9.** Operational policy strategy for developing sorghum in Wonogiri Regency, Central Java.

| Operational Policy Strategy | Activity |
|---|---|
| S-O<br>Increasing productivity in accordance with regional agro-climatic conditions to meet the high demand for sorghum-processed products and products | (1) Make site-specific SOP for sorghum cultivation<br>(2) Socialization and request for SOP implementation of site-specific sorghum cultivation to accelerate adoption<br>(3) Preparation of maps of actual and potential land availability for large-scale development<br>(4) Formulation of an action plan for potential land improvement for land improvement and optimization<br>(5) Wide-scale development in accordance with land availability and suitability<br>(6) Introduction of sorghum processing technology |
| S-T<br>Optimizing cultivation techniques with innovative technology according to regional agro-climatic conditions to overcome pest attacks | (1) Application of site-specific SOP for sorghum cultivation<br>(2) Integrated pest control technical guidance<br>(3) Assistance and escort by field extension officers |
| W-O<br>Collaboration with off-takers related to guaranteeing reasonable prices to increase production to meet the high demand for sorghum-processed products and products | (1) Making an MoU with the off-taker regarding occupation and a reasonable price<br>(2) Introduction of primary and secondary processing of sorghum mechanization<br>(3) Technology introduction and training on sorghum processing to increase added value<br>(4) Improvement of sorghum business institutional governance involving the government, private sector, and farmers<br>(5) Market expansion |
| W-T<br>Collaboration with off-takers regarding guaranteed reasonable prices to increase income so that farmers have the financial ability to deal with pest attacks | (1) Making an MoU with the off-taker regarding occupation and a reasonable price |

Noted: Source = Primary Data Analysis, 2022.

**Table 10.** Operational policy strategy for developing sorghum in Gunungkidul Regency, Yogyakarta.

| Operational Policy Strategy | Activity |
|---|---|
| S-O<br>Take advantage of the land's potential with technology in increasing productivity and product quality to meet high market demand and prices that are starting to improve | (1) Mapping actual and potential land suitability<br>(2) Applying upstream cultivation technology by using high-yielding varieties and improving the cropping pattern<br>(3) Application of processing technology for sorghum products according to SOP<br>(4) Business meetings with partners and consumers<br>(5) Doing MOU between farmer producers and off-takers<br>(6) Application of an intercropping system with other plants<br>(7) Selection of appropriate food or horticultural crops and having high economic value |
| S-T<br>Optimizing land in the third growing season and other fields to reduce competition in land use for other food crops | (1) Application of an intercropping system with other plants<br>(2) Selection of appropriate food or horticultural crops and having high economic value |
| W-O<br>Improving more intensive cultivation of sorghum to meet market demands and improve prices | (1) Technical guidance on sorghum cultivation<br>(2) Technical guidance on processing sorghum products and their derivatives |
| W-T<br>Improving sorghum cultivation to reduce land use competition | (1) Application of intercropping and rotational intercropping technology<br>(2) Use of early maturing varieties for the production of sorghum for food and sugarcane for livestock |

Noted: Source = Primary Data Analysis, 2022.

## 5. Conclusions

Sorghum farming is feasible to be developed in Wonogiri Regency, Central Java, and Gunungkidul Regency, Yogyakarta because it provides a profit value greater than production costs with a BCR value of >1. The perception of farmers in Central Java regarding the development of sorghum is included in the very good category with an average value of 3.31, and the perception of farmers in Yogyakarta is included in the good category with an average value of 2.55. The priority strategy for developing sorghum in Wonogiri Central Java and Gunungkidul Yogyakarta is the expansion strategy (S-O). The operational policy strategy for developing sorghum in Wonogiri Central Java is: (1) Increasing productivity in accordance with regional agroclimatic conditions to meet the demand for high yields and processed sorghum products (S-O); (2) Optimizing cultivation techniques with innovative technologies according to regional agro-climatic conditions to overcome pest attacks (S-T); (3) Collaboration with off-takers related to guaranteeing reasonable prices to increase production in order to meet the high demand for sorghum processed products and products (W-O); and (4) Collaboration with off-takers related to reasonable price guarantees to increase income, so that farmers have the financial ability to deal with pest attacks (W-T). The operational policy strategy for developing sorghum in Gunungkidul Yogyakarta is: (1) Utilizing the potential of land with technology in increasing productivity and product quality and establishing partners (S-O); (2) Optimizing land in the third planting season and other land to reduce competition in land use for other food crops (S-T); (3) Improving more intensive sorghum cultivation techniques to meet market demand and improve prices (W-O); and (4) Improving sorghum cultivation technology to reduce land use competition (W-T).

**Author Contributions:** Conceptualization, S.W. (Sugeng Widodo), J.T., D.S., A.B.P., K., H.P., F.D.A., S., S.W. (Setyorini Widyayanti) and R.H.P.; methodology, S.W. (Sugeng Widodo), J.T., D.S., A.B.P., K., H.P., F.D.A., R.H.P., A.S.R., S., S.W. (Setyorini Widyayanti), A.Y.F. and M.; software, J.T., D.S., R.H.P. and A.S.R.; validation, S.W. (Sugeng Widodo), J.T., D.S., A.B.P., K., H.P., F.D.A., R.H.P., A.S.R., S., S.W. (Setyorini Widyayanti), A.Y.F. and M.; formal analysis, J.T., D.S., R.H.P. and A.S.R.; investigation, S.W. (Sugeng Widodo), J.T., D.S., A.B.P., K., H.P., F.D.A., R.H.P., A.S.R., S., S.W. (Setyorini Widyayanti), A.Y.F. and M.; resources, S.W., J.T., D.S., A.B.P., K., H.P., F.D.A., R.H.P., A.S.R., S., S.W. (Setyorini Widyayanti), A.Y.F. and M.; data curation, J.T., D.S., A.B.P., K., H.P., F.D.A., R.H.P., A.S.R., S., S.W. (Setyorini Widyayanti), A.Y.F. and M.; writing—original draft preparation, S.W., J.T., D.S., A.B.P., K., H.P., R.H.P., S., S.W. (Setyorini Widyayanti), A.S.R., A.Y.F. and M.; writing—review and editing, S.W., J.T., D.S., A.B.P., K., H.P., F.D.A., R.H.P., S., S.W. (Setyorini Widyayanti), A.S.R., A.Y.F. and M.; visualization, S.W. (Sugeng Widodo), J.T., D.S., R.H.P. and A.S.R.; supervision, A.Y.F. and M.; project administration, A.B.P., F.D.A., S.W. (Setyorini Widyayanti) and A.S.R.; funding ac-quisition, S.W. (Sugeng Widodo). All authors have read and agreed to the published version of the manuscript.

**Funding:** This research was funded by the National Research and Innovation Agency of Indonesia through Research Organization for Governance, Economy, and Community Welfare through a research project entitled Prospects and Potential for Sorghum Development to Support Self-Reliance in Food, Animal Feed and Energy in the Central Area of Yogyakarta and Central Java. Grant number B-1832/III.12/PR.03.08/9/2022.

**Institutional Review Board Statement:** Not applicable.

**Data Availability Statement:** The data presented in this study are available upon request from the corresponding author. The data are not publicly available yet but will be in due course.

**Acknowledgments:** The authors express their gratitude to the National Research and Innovation Agency of Indonesia and thanks to Sugeng Hariyadi and Rismiyadi who have helped in collecting data in the field.

**Conflicts of Interest:** The authors declare no conflict of interest.

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
