# Peer review of "Economic Value, Farmers Perception, and Strategic Development of Sorghum in Central Java and Yogyakarta, Indonesia"

_agriculture, doi:10.3390/agriculture13030516_

Round 1

Reviewer 1 Report

As people diversify their diets and the proportion and demand for sorghum and other miscellaneous grains for consumption increases, it is relevant and valuable for the authors to analyse the economic value, farmer perceptions and strategic development of Sorghum in Central Java and Yogyakarta, Indonesia. However, there are still parts that need to be revised.

1. It is suggested to update the literature, sort out and summarize the important literature related to the research topic.

2. In Subsection “2.1. Research Site”, The authors need to justify the selection of Central Java and Yogyakarta, Indonesia as the study sites for sorghum and illustrate the position of these two regions in sorghum production using relevant statistics (such as the share of sorghum production or area planted in these two regions in the national total) to illustrate their representativeness.

3. Are the input and output data in Table 2 derived from research data? If so, please indicate the source of the data below the table.

4. In Subsection “3.2. Farmers Perception” Line 243-247, the author needs to explain why the difference in the views of farmers in these two regions has arisen? What are the main reasons for this result?

5. The authors' analysis of the SWOT of the two regions in terms of sorghum development in section 3.3 and section 4.3 needs to be supported by relevant statistics to support some of the interpretations, such as the analysis of internal, external and priority key factors in section 3.3.1 Line 243-247.

6. The measurements and results of farmers perception needs more evidence. Please give the detailed explanation. The authors can tabulate the measures and statistical differences in the perceptions of farmers in the two regions in section “4.2. Farmers Perception”.

7. Paper language needs to be polish and simplified. It should be improved by a native English speaker.

Author Response

Dear Reviewer 1,

We are happy to send the first revision of manuscript entitled “Economic Value, Farmer Perceptions and Strategic Development of Sorghum in Central Java and Yogyakarta, Indonesia” to be considered for publication in Agriculture Journal. We thank you for your valuable comments/suggestions allowing us to further improving the quality of our manuscript for the first round of review. We have addressed your comments/suggestions in the revised manuscript.

We summarized the revisions as follows: 

1. It is suggested to update the literature, sort out and summarize the important literature related to the research topic.

Author: Thank you for the suggestion. We have reduced some of the less relevant literature and replaced some of the more relevant recent literature. We have adjusted the reduction and replacement of literature between the literature in the text and in the References.

2. In Subsection “2.1. Research Site”, The authors need to justify the selection of Central Java and Yogyakarta, Indonesia as the study sites for sorghum and illustrate the position of these two regions in sorghum production using relevant statistics (such as the share of sorghum production or area planted in these two regions in the national total) to illustrate their representativeness.

Author: Thank you for the suggestion. We have confirmed with the Central Bureau of Statistics for both districts, but did not obtain data on the area of ​​sorghum development in the two study locations, because sorghum is not a priority commodity. Sorghum is an alternative food commodity to replace wheat at this time with restrictions on wheat exports from wheat producing countries. As information that the two locations are centers of sorghum development, we have conveyed this in the Introduction section line 64-68 and in section 2.1. Research Site line 187-200.

3. Are the input and output data in Table 2 derived from research data? If so, please indicate the source of the data below the table.

Author: Thank you for the suggestion. The data that we present in Table 2 is the primary data from the survey results that we have processed. We have added a description of the data sources below Table 2.

4. In Subsection “3.2. Farmers Perception” Line 243-247, the author needs to explain why the difference in the views of farmers in these two regions has arisen? What are the main reasons for this result?

Author: Thank you for the suggestion. We have added an explanation of the differences in farmers' perceptions of sorghum development in the two regions with the percentage of respondents in each perception criterion. We present these improvements in the text of the line 391-394.

5. The authors' analysis of the SWOT of the two regions in terms of sorghum development in section 3.3 and section 4.3 needs to be supported by relevant statistics to support some of the interpretations, such as the analysis of internal, external and priority key factors in section 3.3.1 Line 243-247.

Author: Thank you for the suggestion. We have added an explanation of how to rank the priorities of internal and external factors at the SWOT analysis stage. We present these improvements in section 2.4.3. line 327-331. We have added the results of prioritizing internal and external factors in percent in Tables 3 and 4.

6. The measurements and results of farmers perception needs more evidence. Please give the detailed explanation. The authors can tabulate the measures and statistical differences in the perceptions of farmers in the two regions in section “4.2. Farmers Perception”.

Author: Thank you for the suggestion. We have added evidence of differences in farmer perceptions in the two locations by including the perceived value of the indicators that made the difference between the two locations. We have presented this explanation in the line 509-520.

7. Paper language needs to be polish and simplified. It should be improved by a native English speaker.

Author: Thank you for the suggestion. We've been trying to improve the paper language.

Thank you for your consideration of this manuscript.

Sincerely,

Anggi Sahru Romdon

Reviewer 2 Report

Economic Value, Farmers Perception and Strategic Development of Sorghum in Central Java and Yogyakarta, Indonesia

Thank you for sharing the interesting paper on “Economic Value, Farmers Perception and Strategic Development of Sorghum in Central Java and Yogyakarta, Indonesia”. This article deals with an important subject, which is to evaluate the economic value, farmers' perceptions and specific strategies for developing sorghum in Central Java and Yogyakarta, Indonesia. The study used survey data of 120 respondents and three models such as benefit analysis, scoring technique, and SWOT analysis. The study finds that sorghum farming is feasible to develop in Wonogiri Central Java and Gunungkidul Yogyakarta because it provides a profit value greater than production costs with a BCR value of >1. The perception of farmers in Central Java regarding the development of sorghum is included in the very good category with an average value of 3.31, and the perception of farmers in Yogyakarta is included in the good category

with an average value of 2.55.

Although the paper has positives, a few concerns should be addressed.

·       Detail explanation of sampling design is required.  

·       Typo errors are observed in the manuscript, such as in section 4.3.

·       Abbreviated word should be provided in full form in the first appearance. 

·       Descriptive statistics is missing. Therefore, it is not clear from the study about the variables used in the study.

·       Data Analysis section 2.3.1 should provide a bit more details to understand the threshold value of B/C to say better economic value. Similar explanation is expected for section 2.3.2.

·       How variables and models selected is missing. Author(s) are suggested to detail the variables and model selection over other existing models supporting literature.

·       SWOT analysis results seems to be some expected suggestions from the author/s. It would be better to have number (%) of respondents supported or agreed on the points in SWOT analysis tables.

·       The study should highlight both limitations and future directions for research.

Author Response

Dear Reviewer 2,

We are happy to send the first revision of manuscript entitled “Economic Value, Farmer Perceptions and Strategic Development of Sorghum in Central Java and Yogyakarta, Indonesia” to be considered for publication in Agriculture Journal. We thank you for your valuable comments/suggestions allowing us to further improving the quality of our manuscript for the first round of review. We have addressed your comments/suggestions in the revised manuscript.

We summarized the revisions as follows:

  • Detail explanation of sampling design is required.  

Author: Thank you for the suggestion. We have added a new section 2.2. Sampling Design from line 220-224.

  • Typo errors are observed in the manuscript, such as in section 4.3.

Author: Thank you for the suggestion. We've fixed a typo in section 4.3.

  • Abbreviated word should be provided in full form in the first appearance. 

Author: Thank you for the suggestion. We've fixed the abbreviation at the beginning of the sentence.

  • Descriptive statistics is missing. Therefore, it is not clear from the study about the variables used in the study.

Author: Thank you for the suggestion. This research does not use descriptive statistics, but describes indicators to see perceptions. We present this explanation in section 2.3. Data collection from line 243-251.

  • Data Analysis section 2.3.1 should provide a bit more details to understand the threshold value of B/C to say better economic value. Similar explanation is expected for section 2.3.2.

Author: Thank you for the suggestion. We have added BCR value thresholds that provide benefits to farmers and those that do not provide benefits with the support of the literature. We present these additions in section 2.4.1. from line 262-262. We have also added the criteria for each perception in section 2.4.2. from line 287-289.

  • How variables and models selected is missing. Author(s) are suggested to detail the variables and model selection over other existing models supporting literature.

Author: Thank you for the suggestion. This research does not use a model, but uses a descriptive method for farmers' perceptions and financial analysis for economic value. As support for knowing perceptions, indicators were used in the interview. We present this explanation in section 2.3. Data Collection from line 243-251.

  • SWOT analysis results seems to be some expected suggestions from the author/s. It would be better to have number (%) of respondents supported or agreed on the points in SWOT analysis tables.

Author: Thank you for the suggestion. We have added an explanation of how to rank the priorities of internal and external factors at the SWOT analysis stage. We present these improvements in section 2.4.3. line 327-331. We have added the results of prioritizing internal and external factors in percent in Tables 3 and 4.

  • The study should highlight both limitations and future directions for research.

Author: Thank you for the suggestion. We have added both limitations and future directions for research in section 4.3. from line 833-838.

Thank you for your consideration of this manuscript.

Sincerely,

Anggi Sahru Romdon

Reviewer 3 Report

Dear authors,

I find the basic idea of the study very interesting, but probably because of the desire to cover as many topics as possible, the description of methods and data analysis is brief (at least in the case of economic perception and analysis). I encourage you to:

1. Check the authors' names carefully.

2. What is the situation of farmers in these regions, basic activities, number of farmers, differences of any kind of these regions. I can't figure out how large or small this group of farmers is or the areas available for growing different crops (e.g. in terms of infrastructure)?

3. Why is there a difference (in numbers) between the selected respondents? The collection of this data seems to me to be under-described, as without a centralisation of the number of respondents, according to their role, there may be differences that can compromise the result. Also, which were the selection criteria for respondents? Why 70 and not 60 (example)?

4. I can't figure out what were the topics (questions) set in the interviews. These should be described as accurately as possible.

5. As far as I understand the specific strategies on the basis of which the SWOT analysis was done, are documents prepared by other institutions. In this respect the sources for each finding highlighted should be mentioned.

6. Research hypotheses are not highlighted

7. Each table and figure should note the source of the data!

8. Figure 3 seems to me insufficiently analysed, considering the importance you give it in the study.

In conclusion, the data collected and used must be defined in a clear and detailed manner (materials and methods section) in order to better understand the steps taken and the achievement of the objectives set, as I cannot comment on the basis of the economic analysis and the perception of sorghum growers.

Author Response

Dear Reviewer 3,

We are happy to send the first revision of manuscript entitled “Economic Value, Farmer Perceptions and Strategic Development of Sorghum in Central Java and Yogyakarta, Indonesia” to be considered for publication in Agriculture Journal. We thank you for your valuable comments/suggestions allowing us to further improving the quality of our manuscript for the first round of review. We have addressed your comments/suggestions in the revised manuscript.

We summarized the revisions as follows:

  1. Check the authors' names carefully.

Author: Thank you for the suggestion. We have fixed the author’s name.

  1. What is the situation of farmers in these regions, basic activities, number of farmers, differences of any kind of these regions. I can't figure out how large or small this group of farmers is or the areas available for growing different crops (e.g. in terms of infrastructure)?

Author: Thank you for the suggestion. We have added the characteristics of land resources and farmer activities in the two research locations. We have presented the improvements in section 2.1. Research Site from line 190-200.

  1. Why is there a difference (in numbers) between the selected respondents? The collection of this data seems to me to be under-described, as without a centralisation of the number of respondents, according to their role, there may be differences that can compromise the result. Also, which were the selection criteria for respondents? Why 70 and not 60 (example)?

Author: Thank you for the suggestion. We have added an explanation of how to determine the number of respondents and the criteria. We have presented this explanation in section 2.3. Data Collection from line 230-238.

  1. I can't figure out what were the topics (questions) set in the interviews. These should be described as accurately as possible.

Author: Thank you for the suggestion. We have added the topic set used in the interview. We have presented the topic set in section 2.3. Data Collection from line 243-251.

  1. As far as I understand the specific strategies on the basis of which the SWOT analysis was done, are documents prepared by other institutions. In this respect the sources for each finding highlighted should be mentioned.

Author: Thank you for the suggestion. We have added information that during the FGD, each participant delivered material according to their field and activities related to sorghum development, as material for discussion on determining priority factors for developing a sorghum development strategy. We present this explanation in section 2.3. Data Collection from line 327-331.

  1. Research hypotheses are not highlighted

Author: Thank you for the suggestion. We have added a research hypothesis in the Induction from line section 176-177.

  1. Each table and figure should note the source of the data!

Author: Thank you for the suggestion. We've added data source descriptions to each Table.

  1. Figure 3 seems to me insufficiently analysed, considering the importance you give it in the study.

Author: Thank you for the suggestion. We have added the explanation of Figure 3 in section 3.2. Farmer's Perception from line 391-394. To strengthen this explanation, we have added the value of the perception criteria in section 2.4.2. from line 287-289.

Thank you for your consideration of this manuscript.

Sincerely,

Anggi Sahru Romdon

Round 2

Reviewer 1 Report

The author has made changes to the comments I made and suggests that this paper be published.

Reviewer 3 Report

Dear authors, The manuscript has a very well defined structure and content as a result of the changes that have been made, and all the points raised have been addressed. All the best!